# Effect of Two-Head Flared Hole on Film Cooling Performance over a Flat Plate

**Xuan-Truong Le [1], Duc-Anh Nguyen [1], Cong-Truong Dinh [1,\*] and Quang-Hai Nguyen [2]**

1 Department of Aerospace Engineering, Hanoi University of Science and Technology, No. 1, Dai Co Viet Road, Hai Ba Trung District, Hanoi 100000, Vietnam; truong.lexuan@hust.edu.vn (X.-T.L.); anh.nd150091@sis.hust.edu.vn (D.-A.N.)

2 Viettel Aerospace Institute, Viettel Building—Hoa Lac Hi-Tech Park, Thach That, Hanoi 100000, Vietnam; hainq18@viettel.com.vn

\* Correspondence: truong.dinhcong@hust.edu.vn; Tel.: +84-243-8682525

**Abstract:** Film cooling is commonly utilized in turbine blades to decrease the temperature of the air stream from the combustion chamber that contacts directly with the blades. The shape of a cylindrical hole (CH) with the geometrical variations at inlet and outlet ports was examined using the 3D Reynolds-averaged Navier–Stokes equations (RANS) with a shear stress transport (SST k − ω) turbulence model to study the effect of the two-head flared hole on film cooling effectiveness (FE) at high accuracy with a small y+ value. To assess the effect of the changes, each geometry of the hole was changed one after another while the other parameters were kept invariable at the test value (cylindrical hole). The numerical laterally averaged film cooling effectiveness ($\eta_l$) of the CH case was validated and compared to the experimental data. The simulation results with the two-head flared hole indicated that most of these shape changes increase the FE as compared to the CH case. In particular, the maximum spatially averaged film cooling effectiveness ($\eta_s$) with hole shape expanded along the flow direction at the outlet port reached 60.787% in comparison to the CH case.

**Keywords:** two-head flared hole; film cooling effectiveness; Reynolds-averaged Navier–Stokes analysis

## 1. Introduction

Nowadays, in the gas turbine industry, the engine combustion chamber operates at a very high capacity, creating a large amount of heat discharged into the turbine area. According to Dixon and Hall [1], the material that makes the blades of the turbine cannot withstand this great amount of heat with a maximum temperature of 1000 °C. Therefore, the research on reducing the temperature on the turbine blades is a very active field of development. Turbine blade cooling plays an important role in increasing the thermal operating range of the gas turbine.

To obtain the highest FE, a variety of parameters must be considered; for example, the ratio between the length and diameter of the hole (L/D), the blowing ratio (M), the geometry of the hole, the ratio between the density coolant flow and hot gas, hot gas turbulent intensity, and coolant flow ejection angle. Lutum and Johnson [2] and Burd et al. [3] performed experimental studies on the influence of L/D of the hole on the FE with CH. The experimental results showed that the L/D ratio significantly affected FE. Lee and Kim [4] numerically optimized a CH with the streamwise ejection angle and L/D ratio using the SST turbulence model. $\eta_s$ improved by 3.6% compared to the reference geometry. Yuen and Martinez-Botas [5] performed experimental studies on the effects of the ejection angle with streamwise angles of 30°, 60°, and 90° on the FE for single jets. They reported that the single 60° and 90° jets generated a smaller coefficient of heat transfer than the 30° jet.

Park et al. [6] presented the control of vortex flows by using forward and backward injection jets with various vortex intensities and hole dispositions to increase FE. Li and Zhang [7] numerically analyzed the effect of the geometry of the hole on the FE of a hole

under trunk-branch shape with various M. The results showed that the FE is improved at M = 0.5. Li et al. [8] numerically investigated the effect of the aperture ratios and injection angles of sister-shaped holes on FE. The results revealed that the FE was higher than that of the CH case at all M. Yao and Zhang [9] conducted numerical and experimental studies on the FE on a flat plate with a converged slot-holes row. The results showed that the value of the heat transfer coefficient for the console case was slightly higher than that of the CH case.

Hyams and Leylek [10] numerically presented a detailed assessment of FE with various M, density ratio (DR), and L/D. The results revealed that the laterally diffused hole and simple angle hole made the best FE. Saumweber et al. [11] presented a study of the freestream turbulence influences on the FE with a variety of hole types. The results revealed that the FE is decreased with an increase in turbulence intensity at low M for CH. Silieti et al. [12] presented the numerical research on the FE of gas turbine blades with three turbulence models: SST, realizable k-$\varepsilon$, and v$^2$-f models. The results revealed that hybrid and hexahedral grids are the same in FE. Gritsch et al. [13] have carried out experimental research to assess the effect of fan-shaped hole variations on thermal efficiency. Among these parameters, most hole shapes only make a weak influence on FE. Goldstein et al. [14] presented the influences of the hole's shape and density on 3-D cooling efficiency using an experimental technique. The results showed that the FE was considerably increased with the shaped cooling holes. Liu et al. [15] simultaneously carried out numerical simulations and experimental measurements for the FE of slot holes with waist shape. The FE of slot holes with waist shape was similar to that of the CH with a big divergence angle and the heat transfer coefficient in the zone near the hole centerline was higher to that in the downstream midspan zone. Yu et al. [16] presented a study concentrating on the influences of diffusion hole-shape on the FE with three holes: shape A: CH, shape B: CH with a 10° forward diffusion, and shape C: same as shape B and a 10° lateral diffusion. The experimental results showed that both the FE and h of shape C were the highest among the tested shapes. Kim et al. [17] experimentally analyzed the hole shape's effects on the FE with CH, two laidback holes, and two tear-drop holes. According to the results, the CH case had a weak FE as compared to the shaped-hole cases, and the laidback hole had the best FE. Fu et al. [18] investigated the effects of the inclination and diffusion angles of the chevron hole. The results revealed that a big inclination angle reduced the FE at M = 1.0 and 1.5, whereas the FE increased with the great diffusion angle at high M = 2.0. Liu et al. [19] performed numerical simulations to examine the FE for a turbine inlet guide vane using fan-shaped holes. The results found that the FE increased significantly with the fan-shaped holes around the leading edge and up to 40% coolant mass flow was saved as compared to those of the CH case.

Kim et al. [20] numerically analyzed the effect of converged-inlet hole shapes with the injection angle of coolant flow, streamwise and expanded angles in streamwise and pitchwise directions, and L/D. It was concluded that the FE increased by 46.5% in comparing with the CH case. Moreover, $\eta_s$ reached the maximum at an injection angle of 40°. In the next research paper, Kim et al. [21] continued the numerical analysis for a converging–diverging hole. With the diverged hole combination, the highest FE increased by 9.9% at M = 1.5 when compared to the fan-shaped hole case. For more complex hole shapes, Kim et al. [22] also studied the FE of four shaped holes: louver, fan-shaped, dumbbell-shaped and crescent. The numerical results indicated that the dumbbell-shaped hole had the highest FE among all tested cases with M from 0.5 to 2. Kohli and Thole [23] analyzed the influences of a cooling hole with an entrance angle of 35° in regard to the main-stream and an exit angle of 15°. Numerical results indicated that the FE was reduced with the orientation of the coolant flow. Gritsch et al. [24] presented a comparison of the FE for the CH and two expanded holes. The experimental results indicated that both expanded holes in the exit part increased the FE, especially at a high M. Hay et al. [25] experimentally studied the inclined holes at an injection angle of 30°. The results reported that the discharge coefficients at the hole inlet rounding was enhanced up to 15%.

An experiment from Afzal et al. [26] studied the distance between the grooves on the plain tubes to maximize the Nusselt number. The study revealed that the Nusselt number will rise with a reduction in the tube surface temperature. Zhang et al. [27] presented an unsteady research on the FE of a rotational turbine blade. The numerical results indicated that the main-stream swing had an impact on the FE and a higher FE in spanwise direction was achieved at a higher swing frequency. Du et al. [28] presented a parametric study of a trenched-shaped hole on FE. The results indicated that the FE at the suction surface of the turbine blades in downstream zones was slightly changed when the trenched holes were modified.

From the above literature review, the shaped holes generally produce a high FE. Forward expanded holes have higher values of FE and lower values of heat transfer coefficient than that of the CH case. However, many of them are difficult to be manufactured. In addition, there are few studies that investigate in detail the expansion of the CH into converged and diverged holes or a combination of them. For that reason, this study proposes a design of a two-head flared hole to evaluate the FE between the hot and cool flows on a flat plate.

## 2. Numerical Analysis

### 2.1. Description of Geometry

Due to its convenience in fabrication and machining, a usual CH is widely used to increase the FE of turbine blades. Therefore, Lee and Kim [4] presented an optimization technique for CH to maximize FE. In this research, the predicted the FE of the CH was compared to the test data of Saumweber et al. [11]. Figure 1 presents the geometry and computational domain, where a CH, extended on inlet and outlet surface holes in streamwise and pitchwise directions, was used to evaluate the FE in this study. The computational domain consists of three main parts: a principal plenum of the hot gas flow, a secondary plenum of the cool flow, and a CH connecting these two flows. The diameter of the hole (D) and the width of both channels are 5 mm and 20 mm, respectively. The ratio between the hole length and hole diameter (L/D) is 6, the injection angle (θ) is 30° and some other parameters are shown in Figure 2.

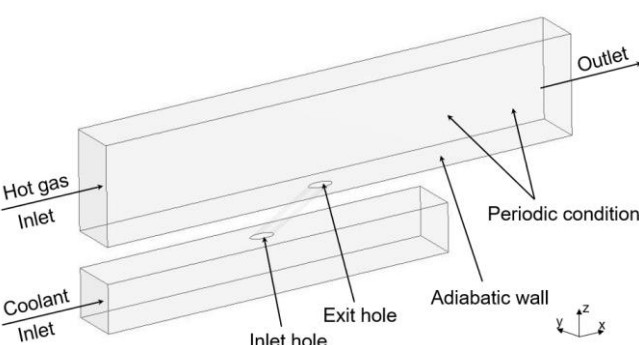

**Figure 1.** Geometry and computational domain.

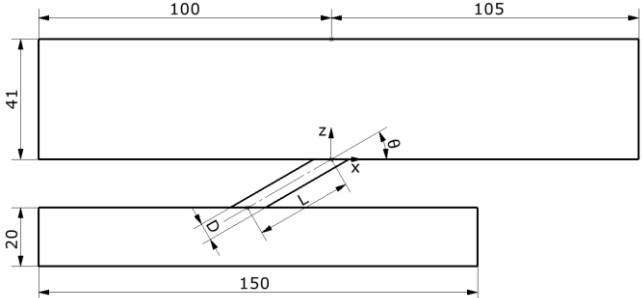

**Figure 2.** Geometrical parameters of CH case.

Figure 3 shows the geometries of a two-head flared hole, where the geometric parameters at the inlet and outlet surface holes are changed and dimensionless with the hole diameter (D). The values of the hole geometric parameters are shown in Table 1. The cross-section transits the two-hole shapes perpendicular to the centerline of the CH at the midpoint of this line (L/2). There are six types of hole shape that need to be investigated, corresponding to three parameters at the inlet surface hole and three parameters at the outlet surface hole. On each inlet and outlet surface hole, the CH was enlarged in three sides, respectively: extend behind (B), expand in front (F) and expand together to the left and right sides (S). To assess the effect of the hole design change, each parameter was changed, respectively, while the other parameters were maintained at a constant reference value.

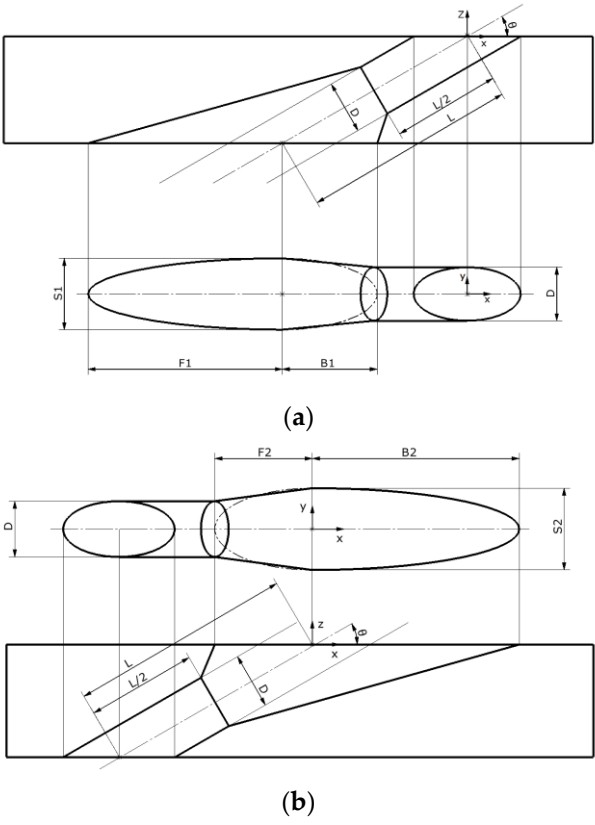

**Figure 3.** Description of studied parameters: (**a**) inlet flared hole, (**b**) outlet flared hole.

**Table 1.** Geometric parameters of two-head flared hole.

| Geometric Parameter | Ref. Value | Lower Bound | Upper Bound | Step |
|:---:|:---:|:---:|:---:|:---:|
| S | 1D | 1.5D | 3D | 0.5D |
| B | 1D | 1.5D | 3D | 0.5D |
| F | 1D | 1.5D | 3D | 0.5D |

The value of $\eta_s$ is defined as:

$$\eta_s\left(\frac{x}{D}, \frac{y}{D}\right) = \frac{\int_0^{21} \int_{-2}^{2} \eta\left(\frac{x}{D}, \frac{y}{D}\right) d\left(\frac{x}{D}\right) d\left(\frac{y}{D}\right)}{21x4}$$

where:

$$\eta\left(\frac{x}{D}, \frac{y}{D}\right) = \frac{T_{aw}\left(\frac{x}{D}, \frac{y}{D}\right) - T_h}{T_c - T_h}$$

$T_{aw}$, $T_h$ and $T_c$ are the temperature on the adiabatic wall, static temperature at the inlet surface of hot gas channel, and coolant temperature at the coolant inlet, respectively. This

performance function was averaged over a surface with a width of 4D and streamwise length of 21D.

### 2.2. Numerical Method

In this study, the commercial software, ANSYS CFX19.1 [29], was employed using 3D RANS with the SST turbulence model to numerically investigate the effect of the two-head flared hole shape on the FE of a flat plate with a mixing of coolant and hot flows. The optimization study of Lee and Kim [4] on the FE of the CH showed that the SST turbulence model has better results than the k-$\varepsilon$ turbulence model in FE. Therefore, in this research paper, the SST turbulence model was selected as a turbulence closure model. The SST turbulence model combines the advantage of k-$\omega$ and k-$\varepsilon$ models with a blending function that makes a smooth transition between two models. The k-$\omega$ and k-$\varepsilon$ models were used for the calculation in the near-wall surface and in the bulk flow region, respectively.

The structured grid system was constructed using ANSYS ICEM for the computational domain. Figure 4a shows that the hexahedral mesh type was used for the CH case with a crowded mesh on the wall region and near the film cooling hole to solve the high-velocity gradients. The mesh in the near-wall zone were refined and the wall spacing on the surface of the main channel and film cooling hole were set to $10^{-3}$ (mm) to guarantee that the value of $y^+$ on these walls is less than 1. The first mesh points adjacent to the wall at other walls were also placed at the value of $y^+ < 2$. The O-grid was applied two times at the film cooling hole for improving the mesh quality, as shown in Figure 4c. Five different grids were investigated to discover the best mesh system, with the numbers of mesh in a range from 600,000 to 2,700,000.

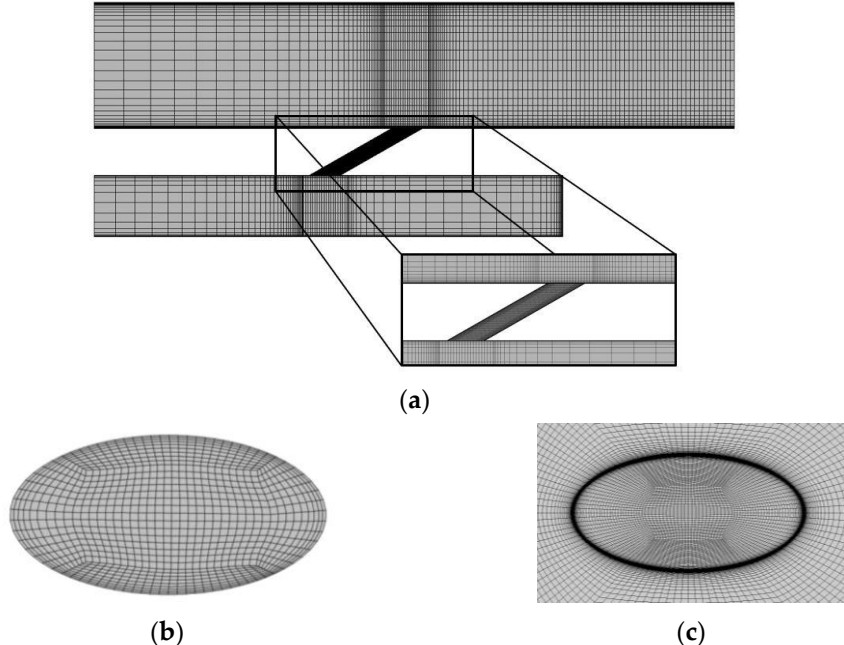

**(a)**

**(b)**                    **(c)**

**Figure 4.** Grid structure and determinant value: (**a**) mesh structure of geometry, (**b**) CH grids of Lee and Kim [9], (**c**) CH grids of present CFD.

The boundary conditions are the same as Lee and Kim [4] and the simulation results were compared with the experimental data performed by Saumweber et al. [11]. The working fluid was chosen as an ideal gas (air). Adiabatic and no-slip conditions were utilized at all the walls, the film cooling hole, the top and bottom surface of the main channel, and the plenum. The periodic conditions were specified at the side boundaries of the main channel. The velocity at the inlet of the main channel, and the static pressure at the outlet of the main channel were, respectively, set to 139.74 m/s and 0 Pa. The temperatures of the hot gas and the coolant were 540° K and 310° K, respectively. At the inlet of the

coolant plenum, the velocity was specified with M = 0.5. The value of DR for the coolant to the main-stream was 1.7, and the turbulence intensity and the turbulence length scale in the main channel were 3.6% and 2.7D, respectively.

For the convergence standard, RMS relative residual values of all flow parameters were set to $10^{-6}$, and the calculation for a single simulation finished within 4000 iterations. Furthermore, the wall temperature and the outlet mass flow rate were monitored to confirm that the convergence is achieved. The computations were performed by an Intel (R) Xeon (R) CPU X5675 @ 3.07 GHz PC. The computation time was typically 18–32 h, depending on the number of mesh and the quality of the meshes.

## 3. Results

### 3.1. Grid Independency Test and Validation

Figure 5 shows the $y^+$ value distributing on the wall surface. The average $y^+$ value is 0.18 and $y^+$ value distribution on all walls is guaranteed to be lower than 1.5, which is appropriate to ensure accurate results with the SST k-ω model. To discover the optimum mesh system, a grid-dependency test was carried out. As presented in Figure 6, five different mesh systems with increasing numbers of nodes from 600,000 to 2,700,000 were evaluated to find the optimum mesh. Figure 6 compares the effect of various grid sizes on the FE at the down wall of the main channel. The results of $\eta_l$ indicate that the number of grids has a notable influence on the results when the number is less than 1 million grids, and when the number of mesh is more than 1.5 million grids, the deviation is negligible.

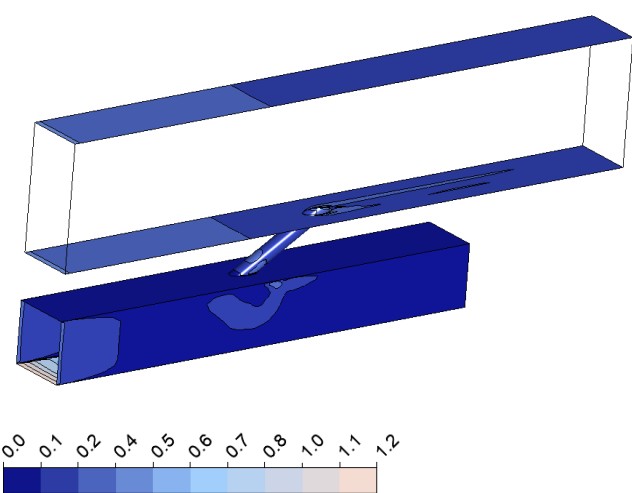

**Figure 5.** $y^+$ contours distribution on the walls.

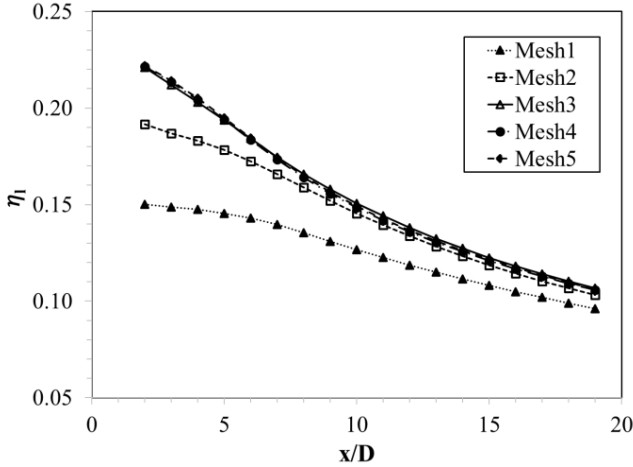

**Figure 6.** Influence of grid mesh on the results.

According to the graph in Figure 6, in the case of 0.6 million grids and 1 million grids, the average deviation percentage between them is very large. However, the average deviation percentage in the case of 2.2 million grids compared to the 1.7 million grids case and the case of the 2.7 million grids compared to the 2.2 million grids case only are 0.56% and 0.46%, respectively. This confirms that the grid converged and ensures stable results in the case of the optimum grid number of about 1.7 million.

To validate the computational results, the obtained data are compared with the measurements from Saumweber et al. [11] and numerical results of Lee and Kim [4] for the same conditions (L/D = 6.0 and $\alpha$ = 30°). As shown in Figure 7, it can be observed that the numerical and experimental results of $\eta_l$ falls with the decrease in x/D. The 1.7 million grid model is used to compare and gives a reasonable agreement with the experiment's data, with the average error percentage being 0.636% and the biggest percentage error being 10.628% at x/D = 19. This is partly because of the uncertainty of the experimental system of Saumweber et al. [11]; with the Mach number being maintained within ±1.6%, uncertainty in M and DR are ±2% and ±1.7%, respectively. However, the average error percentage in this investigation is much smaller than the numerical results of Lee and Kim [4] (9.691%). The better results can be attributed to the improved mesh quality compared to the numerical analysis of Lee and Kim [4], with the mesh near the wall being much better handled, as shown in Figure 4b,c.

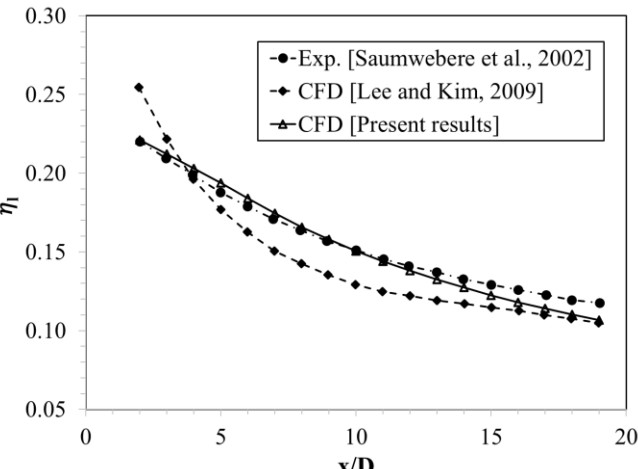

**Figure 7.** Numerical validation.

The velocity contours and vectors on the x-z plane at y/D = 0 for the CH shape are illustrated in Figure 8. With this shape, a large recirculation vortex zone is observed on the lower wall near the hole's inlet. It is this recirculation that causes a sharp rise in velocity close to the hole's upper wall as the coolant flow is pushed out of the outlet into the main channel. This behavior is known as the jetting effect. Caused by an uneven velocity distribution, the large recirculation vortex zone disturbs the flow in the hole and creates a great momentum in the coolant flow at the hole's upper outlet. Such a phenomenon results in the coolant flow spraying strongly into the main-stream, which leads to the coolant flow's detachment from the surface of the main channel and notably decreases the film's cooling effectiveness. To increase the film's cooling performance, the recirculating zone treatment and reduction in the jetting effect is important.

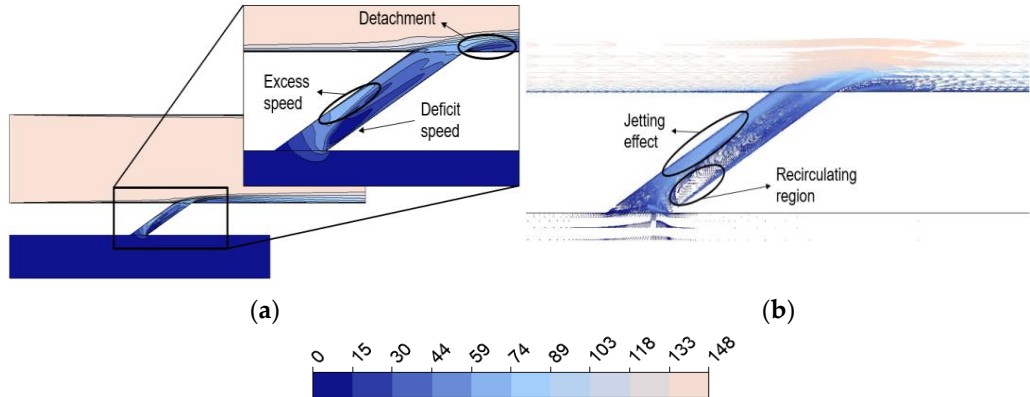

**Figure 8.** Distribution of velocity contours and velocity vectors on x–z plane at y = 0 (unit: m/s): (**a**) velocity contours, (**b**) velocity vectors.

*3.2. Expansion behind the Inlet Hole*

To reduce the jetting effect, expanding behind the inlet hole is a remarkably effective measure. Figure 9 compares the distribution of $\eta_s$ of the CH and the expanded hole cases. The FE is notably improved by using the flared inlet hole shape. With the 1.5D expanded position, $\eta_s$ on the main channel surface increases insignificantly. However, when expanded to the 2D position, the FE reached the maximum and increased by 0.5%, as compared to that of the CH case. In this extended position, the vortices in the hole are also significantly improved, and the recirculating region is broken and markedly reduced. Consequently, the momentum's uniformity at the outlet of this hole is notably enhanced in comparison with that of the reference case, which leads to an increase in FE. As it continues to expand to the 2.5D and 3D position, the FE drops drastically compared to the 2D expanded position. This happens because of the jetting effect caused by the increase in velocity at the cooling hole's neck location (Figure 10). The coolant strongly sprayed into the main-stream also makes the coolant more susceptible to being mixed into the hot stream and decreases the film cooling efficiency.

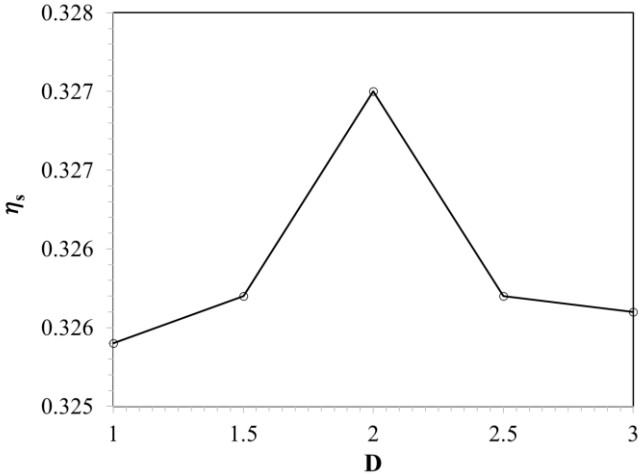

**Figure 9.** $\eta_s$ with expansion behind the inlet hole.

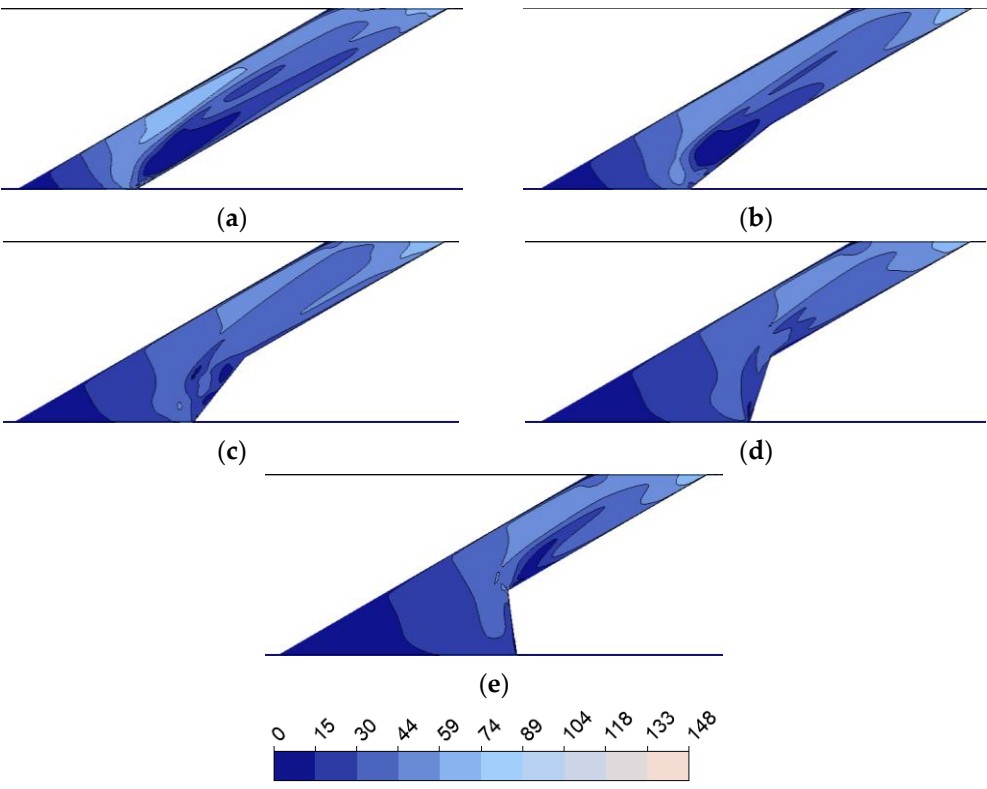

**Figure 10.** Velocity contours on x-z plane at y/D = 0 (unit: m/s): (**a**) reference, (**b**) B1 = 1.5D, (**c**) B1 = 2D, (**d**) B1 = 2.5D, (**e**) B1 = 3D.

### 3.3. Expansion in Front of the Inlet Hole

In Figure 11, the distributions of $\eta_s$ for four inlet hole shapes, which are expanded forward, are compared. The highest FE improvement from the reference case is 1.045% at the 2.5D expanded position. As the shape change of the hole is not big enough, the FE of the 1.5D expanded case is very close to that of the reference design. However, the FE begins to increase at the 2D expanded position and reaches its maximum at the 2.5D expanded position. $\eta_s$ at the 3D expanded position is lightly reduced compared to the 2.5D expanded position. However, the turbine blades consist of different rows of holes, so the expansion of one hole should not affect the others.

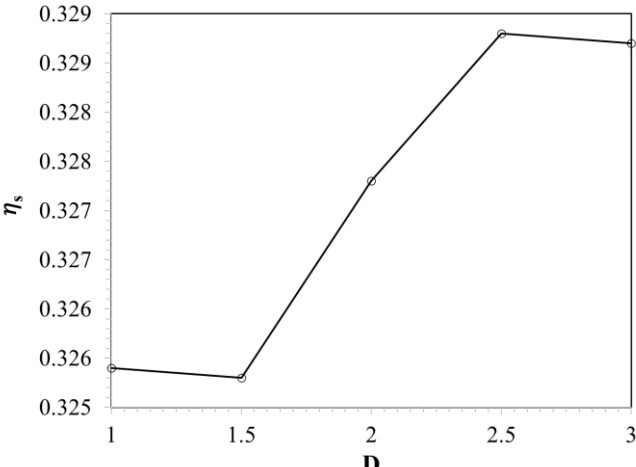

**Figure 11.** $\eta_s$ with expansion in front of the inlet hole.

At the 3D expanded position, it takes up a lot of space for this hole and can be encroached to the position of the front hole affecting blade structure. In order to ensure the blades' structural integrity and achieve the best cooling performance, the case of the 2.5D expanded position will give the best results.

The results of FE distributions on the main channel surface (x-y plane at z/D = 0) are presented in Figure 12. Both the investigated geometries and the reference give similar contours' shape, but the expanded holes show a larger area of high FE. Especially, the cases of the 2.5D and 3D expanded position indicate the largest area of high FE among the investigated cases.

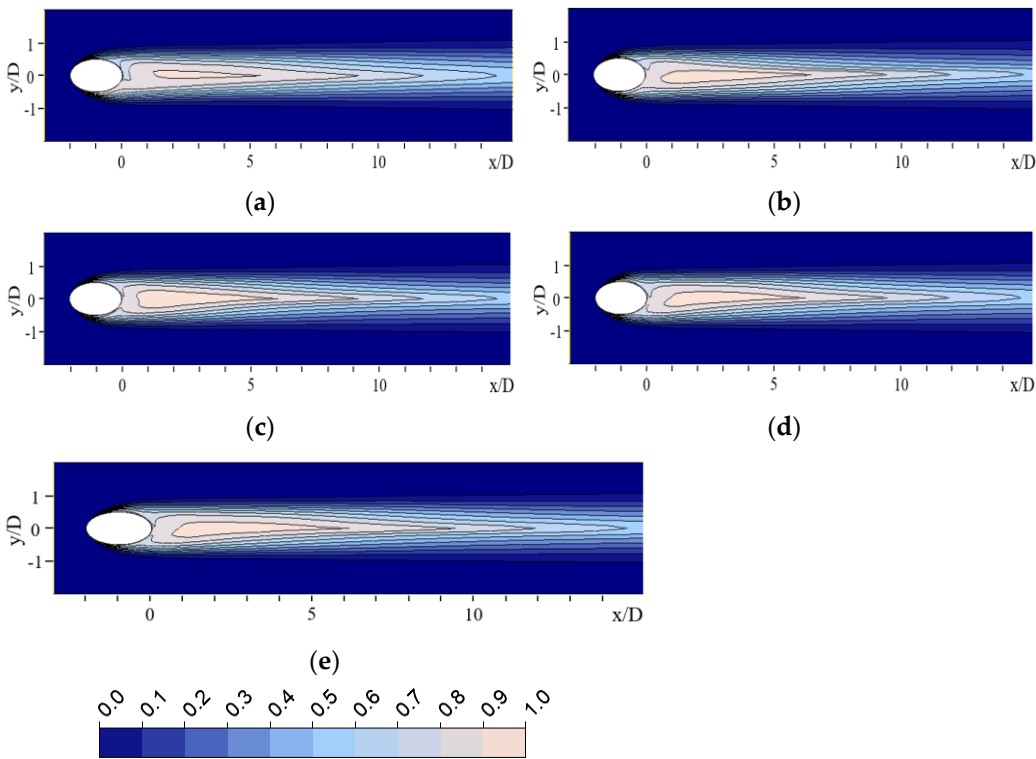

**Figure 12.** FE distributions on the x-y plane at z/D = 0: (**a**) reference, (**b**) F1 = 1.5D, (**c**) F1 = 2D, (**d**) F1 = 2.5€, (**e**) F1 = 3D.

### 3.4. Expansion to the Sides of the Inlet Hole

Expanding to the sides always results in a higher cooling performance than expanding forward or behind. The velocity in the cooling hole becomes higher even before discharging out of the exit hole. Figure 13 presents that $\eta_s$ increased in all of the S1 expanded positions, with at least 2.182% at the 1.5D expanded position compared to that at the reference case. The maximum FE begins around the 3D expanded position (Figure 13). At this range, $\eta_s$ is increased by approximately 4.456% at the 3D expanded position and 4.425% at the 2.5D expanded position compared to that at the reference case.

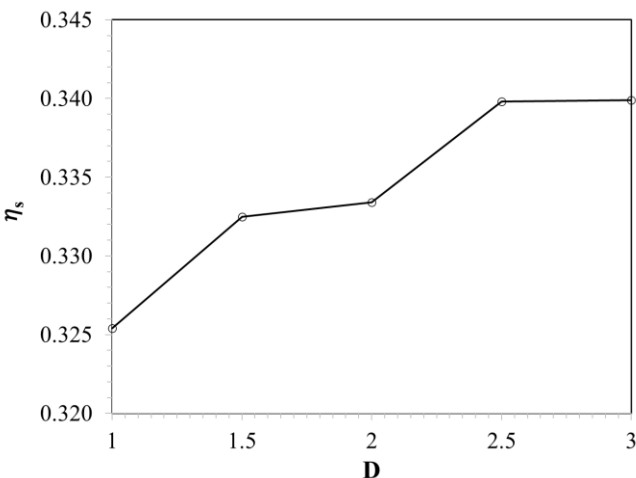

**Figure 13.** $\eta_s$ with expansion to the sides of the inlet hole.

From the velocity contours on the x-z plane at $y/D = 0$ that are shown in Figure 14, the flow in the tube is more stable with the enlargement of the cooling hole. As the exit hole is unchanged, the FE is improved due to a stable flow in the hole. Specifically, the average of the turbulent kinetic energy in the hole of the 2.5D and 3D expanded position cases compared to the reference case is 30.417, 30.618 and 33.232, respectively (unit: J/kg). Therefore, the FE of the flared hole case can only increase to a certain extent. For this reason, in Figure 13, the FE at the 3D expanded position is almost negligible compared to the 2D expanded position.

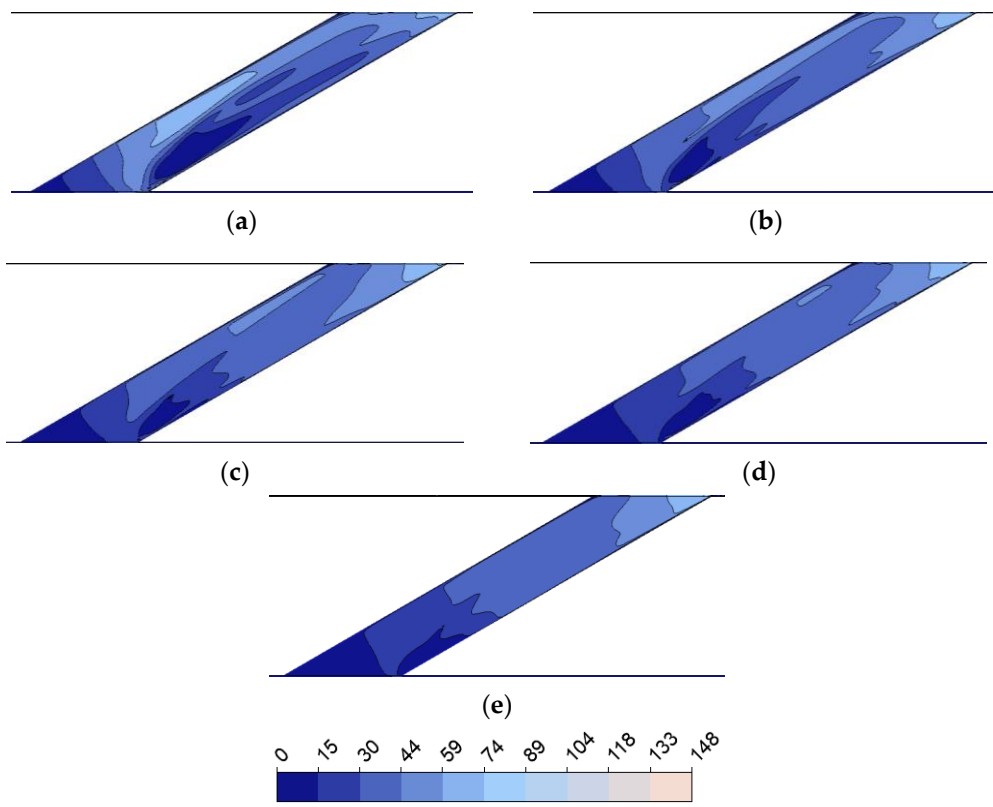

**Figure 14.** Velocity contours on x-z plane at $y/D = 0$ (unit: m/s): (**a**) reference, (**b**) S1 = 1.5D, (**c**) S1 = 2D, (**d**) S1 = 2.5D, (**e**) S1 = 3D.

### 3.5. Expansion behind the Exit Hole

The expansion behind the exit hole markedly increases FE. The expansion in this way will reduce the coolant flow's local velocity at the hole's exit. This reduces the coolant flow's penetration into the hot airstream, avoids the detachment of the coolant from the main channel's bottom surface and significantly improves the film cooling' effectiveness.

From the results of $\eta_s$ in Figure 15, right at the 1.5D expanded position, the performance is sharply increased, as FE gains 6.669% and the highest FE gains 9.926% at the 3D expanded position compared to that of the reference case. $\eta_s$ at the 3D expanded position does not increase significantly compared to the 2.5D expanded position, as FE gains 9.619%. This is because when the expansion is excessively wide, the coolant flow is decelerated considerably so that it will easily mix into the main-stream and only slightly increase the FE (velocity of 35.4 m/s compared to 33.7 m/s, corresponding to the 2.5D and 3D expanded position). In addition, as mentioned above, because the blades of the turbine have a lot of cooling holes, when expanded to the 3D position, the structure of the blade will be greatly altered, reducing the firmness. Therefore, in this case, it is necessary to conduct another investigation on the effect of the turbine blade structure to check the optimal case.

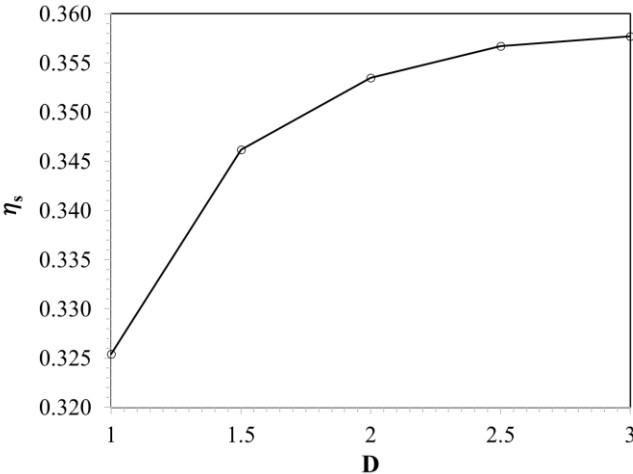

**Figure 15.** $\eta_s$ with expansion behind the exit hole.

From the turbulent kinetic energy contours distributed near the wall on the x-y plane (Figure 16), expanded holes have larger turbulent kinetic energy than the reference hole, and this turbulent kinetic energy is directly proportional to $\eta_s$. This proves that the turbulence increases the interaction of the coolant flow with the wall surface, which boosts heat transfer on the blade surface, increasing the FE. For this reason, the FE of the expanded hole cases is much higher than that of the reference case.

### 3.6. Expansion in Front of the Exit Hole

The forward expansion of the exit hole also reduces the phenomenon of the jetting effect, because when expanded in this way, the coolant flow in the upper wall will have a smaller velocity due to being distributed in a bigger area. However, when the enlargement is too large, there will be one more recirculating region. With the forward expansion of the exit hole, when expanded at a reasonable position, the recirculating region will help control the outflow velocity to avoid the jetting effect. However, in another expanded position, this recirculating region can lead to large turbulences that cause a loss of flow energy, which reduces the film cooling performance.

The value of $\eta_s$ is shown in Figure 17, compared to the reference case; the highest FE is increased by 0.738% at the 2D expanded position. Meanwhile, at the 1.5D expanded position, the effectiveness is reduced by 2.581%. This is because at the beginning of the expansion, the recirculating region at the exit of the expanded hole is not large enough. It

is only a small turbulence flow and it causes the amount of turbulence and vortex in the main-stream to increase, resulting in an uneven and unstable outflow.

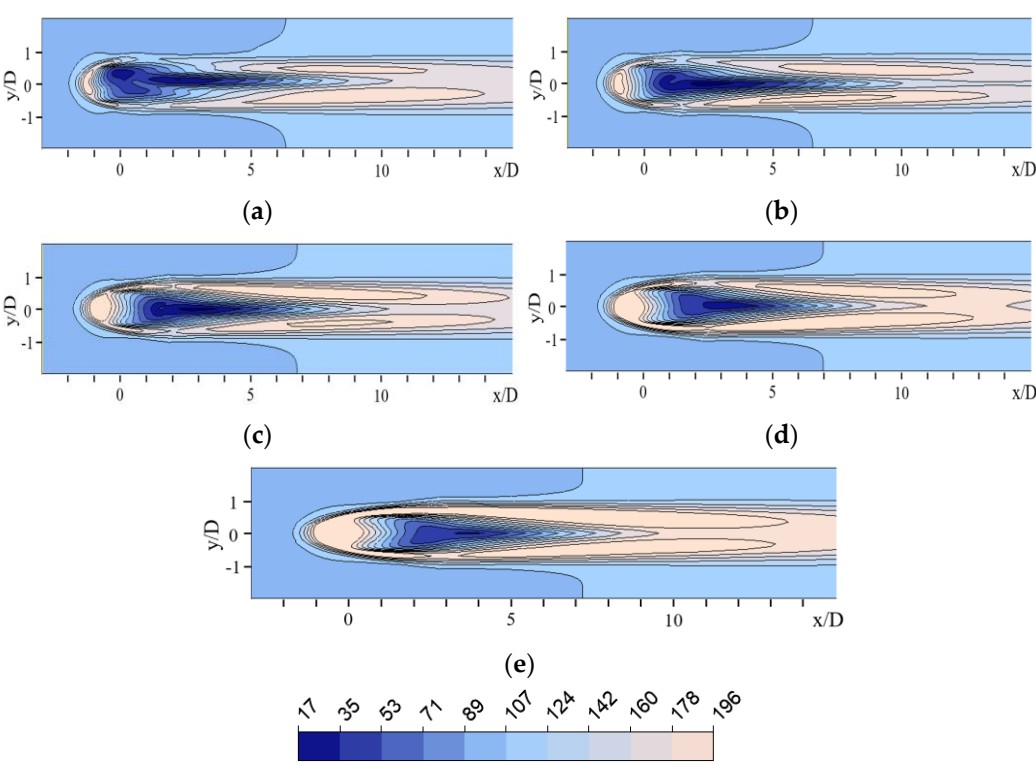

**Figure 16.** Turbulent kinetic energy distributions on the x-y plane at z/D = 0.2 (unit: m²/s²): (**a**) reference, (**b**) B2 = 1.5D, (**c**) B2 = 2D, (**d**) B2 = 2.5D, (**e**) B2 = 3D.

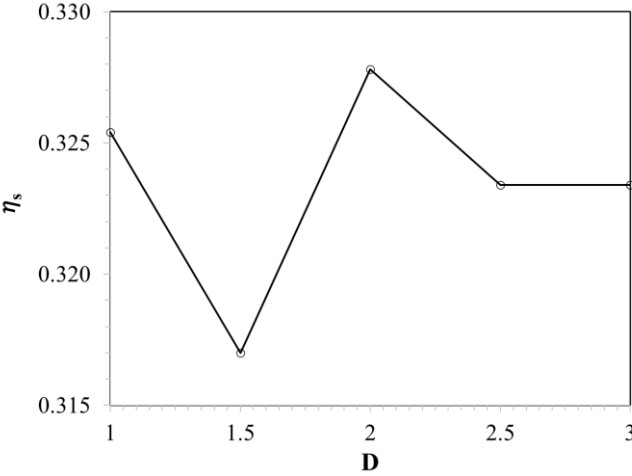

**Figure 17.** $\eta_s$ with expansion in front of the exit hole.

In the 2.5D and 3D expanded cases, the FE is also reduced. In these cases, the flow becomes more complicated due to the formation of an excessively large vortex in the mouth of the extension of the hole, increasing the turbulence at the outlet area and making the coolant flow less stable and more easily mixed into the hot flow (Figure 18). Compared with the case of 2D expansion, the recirculating region, in this case, is sufficient to control and direct the outflow velocity, reducing the jetting effect phenomenon and increasing the FE.

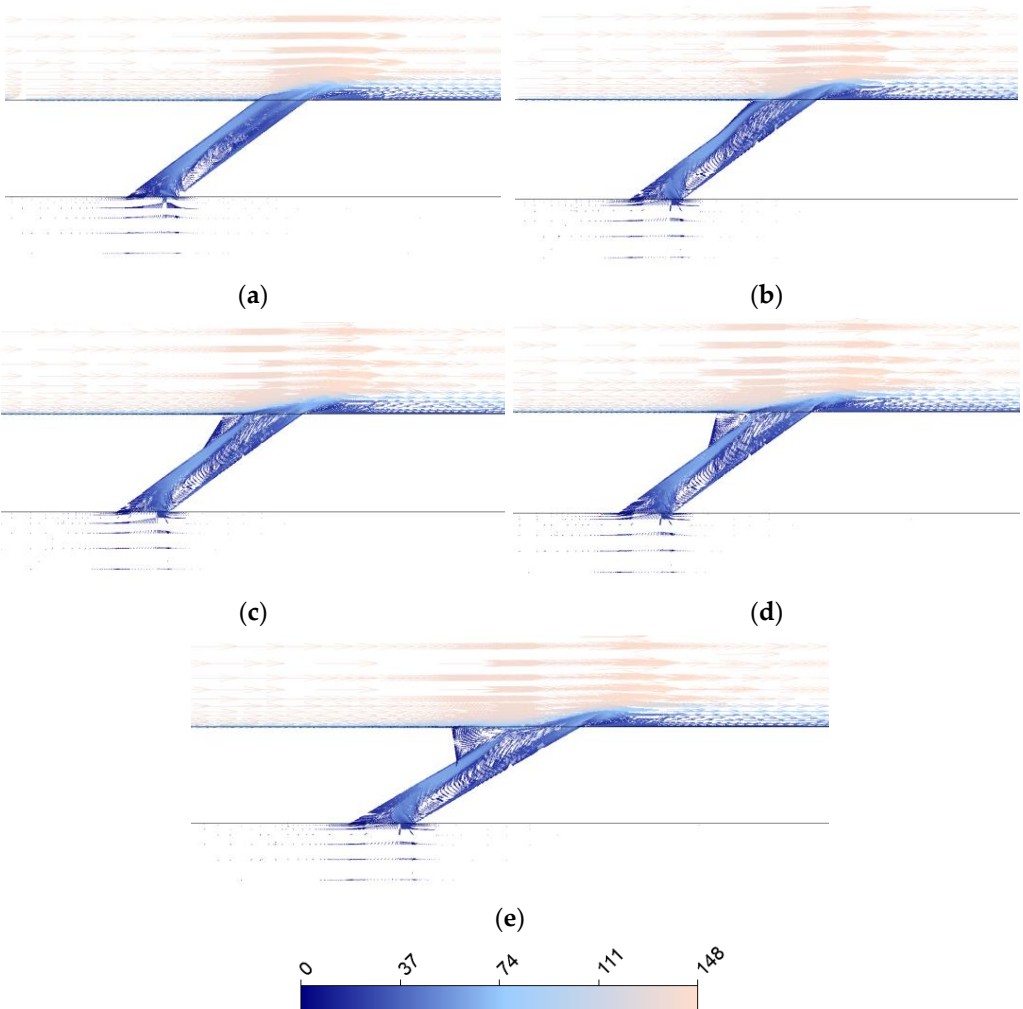

**Figure 18.** Velocity vectors on x-z plane at y/D = 0 [Unit: m/s]: (**a**) reference, (**b**) F2 = 1.5D, (**c**) F2 = 2D, (**d**) F2 = 2.5D, (**e**) F2 = 3D.

*3.7. Expansion to the Sides of the Exit Hole*

This is the expansion measure that brings the highest cooling performance in all of the investigated cases. By expanding in this way, the coolant flow will spread evenly and widely across the surface, causing the coolant flow to cover a large area of the cooling surface. As a result, it avoids the hot flow from contacting the surface and significantly increases the cooling performance.

To prove the effectiveness of the expansion to the sides of the hole, in Figure 19, the distribution of $\eta_l$ and $\eta_s$ of the expanded shapes are compared against that of a CH. These diverged-exit shapes show the large zone of high FE and a superior spreading of the coolant in the streamwise direction in comparison with those of the CH case. Especially, as shown in Figure 19a, a noticeable increase in the FE of about 0.12 to 0.33 is achieved around x/D = 2. In this zone (x/D = 2), the FE is increased by at least 50% for the 1.5D expanded position. It can be seen that the FE quickly reduces as the x/D enhances, especially in the zone of x/D > 10. Figure 19b presents the distributions of $\eta_s$. Right at the 1.5D expanded position, the FE is enhanced by 27.197%, the highest FE is 60.787% at the 3D expanded position. The FE increases steadily at each expanded position. However, at the 3D expanded position, because the hole is too wide, the effectiveness increase is not much compared to the previous expanded case (2D).

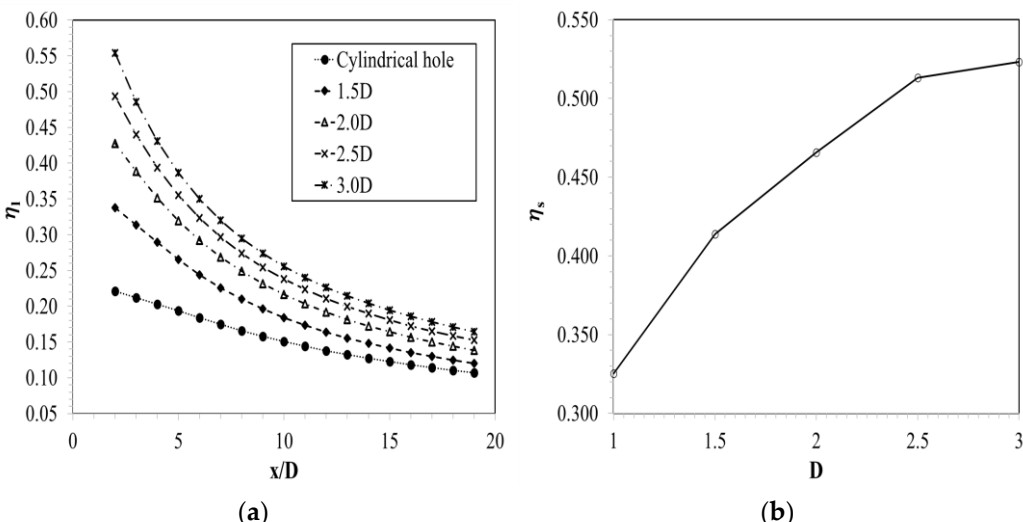

**Figure 19.** FE with expansion to the sides of the exit hole: (**a**) $\eta_l$, (**b**) $\eta_s$.

Figure 20 indicates the local FE distribution contours on the x-y plane at z/D = 0. The cooling performance increases strongly, mainly in the mouth area of the hole. The more the hole is enlarged (S2 increase), the more the effectiveness decreases with increasing x/D, i.e., the cooling performance at the mouth of the hole increases sharply and the effectiveness drops when away from the mouth of the hole. From these results, the distance between the holes can be calculated for the best cover.

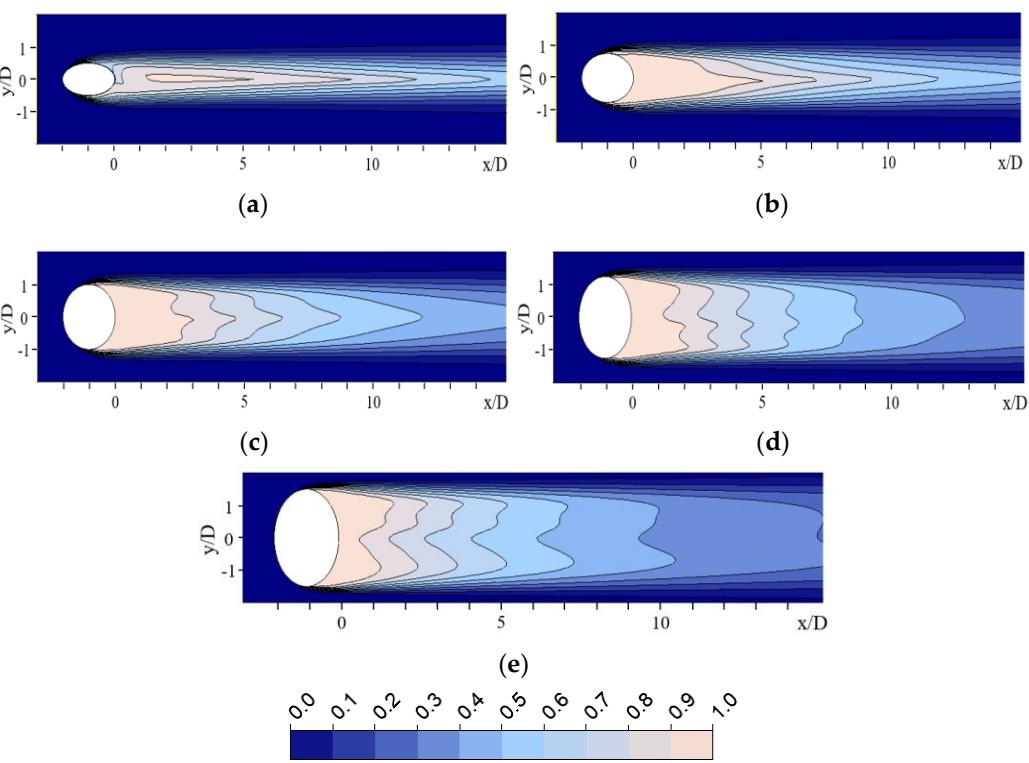

**Figure 20.** FE distributions on the x-y plane at z/D = 0: (**a**) reference, (**b**) S2 = 1.5D, (**c**) S2 = 2D, (**d**) S2 = 2.5D, (**e**) S2 = 3D.

## 4. Conclusions

In this study, a numerical analysis was performed to identify the impacts of the hole's shapes on the film cooling performance on the turbine blades' surface using a converged-

inlet and diverged-outlet shape. The independence of mesh with five meshes was examined and the validation results for the CH case using the SST model closely replicated the results from the experiment.

Based on simulation results, it is confirmed that the two-head flared hole can influence the film cooling performance. The findings are summarized below:

1. For the expansion behind the inlet hole, the highest $\eta_s$ increased by 0.5% at the 2D expanded position. This converged design increased the film cooling's performance by subduing the recirculating region and reducing the jetting effect in the cooling hole.
2. For the expansion in front of the inlet hole, the maximum FE occurred at the 2.5D expanded position, which was 1.045% higher than that of the CH design.
3. In the case of the expansion to the sides of the inlet hole, the highest cooling performance gain was 4.456% at the 3D expanded position. In this case, the velocity field in the pipe became smoother and less turbulent.
4. The expansion behind the exit hole significantly increased the performance. At the 1.5D expanded position, the FE increased by 6.669%, and the highest FE enhanced by 9.926% at the 3D expanded position.
5. A slight increase in performance is also obtained at the forward expansion of the exit hole. The highest FE improved by 0.738% at the 2D expanded position. However, in this way, the cooling performance is not stable.
6. The expansion to the sides of the exit hole provided the best film cooling performance in all six cases of the expanded holes investigated. The FE increased in the range of 27.197% to 60.787%, with expanded positions increasing from 1.5D to 3D, respectively. This is also the optimal film cooling method that many researchers are investigating.

Based on these numerical results, the paper will continue to optimize the design of the two-head flared hole to maximize its performance using the best optimization technique in the future work.

**Author Contributions:** X.-T.L. and Q.-H.N. conceived the idea, D.-A.N. performed the calculations and wrote the paper, C.-T.D. guided and responded to the work. All authors have read and agreed to the published version of the manuscript.

**Funding:** This research is funded by Hanoi University of Science and Technology (HUST) under grant number T2020-PC-012.

**Institutional Review Board Statement:** Not applicable.

**Informed Consent Statement:** Not applicable.

**Data Availability Statement:** The data that support the findings of this work are available within the article.

**Acknowledgments:** The authors would like to express their sincere thanks to the School of Transportation Engineering and Hanoi University of Science and Technology for providing the budget of this project.

**Conflicts of Interest:** The authors declare no conflict of interest.

## Nomenclature

| | | |
|---|---|---|
| B | expand to the front of hole center | (mm) |
| D | diameter of the film-cooling hole | (mm) |
| DR | density ratio, $(\rho_c/\rho_\infty)$ | (-) |
| F | expand to behind hole center | (mm) |
| M | blowing ratio, $(\rho_c U_c)/(\rho_h U_h)$ | (-) |
| S | expand to the sides of hole center | (mm) |
| $T_{aw}$ | adiabatic wall temperature | $(^\circ K)$ |
| $T_h$ | hot gas temperature in the main channel | $(^\circ K)$ |
| $T_c$ | coolant jet temperature | $(^\circ K)$ |
| $U_c$ | velocity of coolant at the hole exit | (m/s) |
| $U_\infty$ | velocity at the main-stream inlet | (m/s) |
| $y^+$ | dimensionless distance | (-) |

**Greek**

| | | |
|---|---|---|
| $\rho$ | density of steam | $(kg/m^3)$ |
| $\eta_s$ | spatially averaged film cooling effectiveness | (-) |
| $\eta_l$ | laterally averaged film cooling effectiveness | (-) |
| $\theta$ | injection angle | $(^\circ)$ |

**Subcripts**

| | |
|---|---|
| aw | adiabatic wall |
| c | coolant |
| h | hot gas |
| l | laterally averaged |
| 1 | at the surface of inlet hole |
| 2 | at the surface of outlet hole |
| $\infty$ | main-stream |

**Acronyms**

| | |
|---|---|
| CH | cylindrical hole |
| FE | film cooling effectiveness |
| RANS | Reynolds-averaged Navier–Stokes |
| SST | shear stress transport |

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
