# Peer review of "Effect of Two-Head Flared Hole on Film Cooling Performance over a Flat Plate"

_aerospace, doi:10.3390/aerospace8050128_

Round 1
Reviewer 1 Report
- 5 sig-figs for FE improvement? I'm not sure that's justified based on the model's assumptions. 60.8% is probably more accurate.
- the nomenclature list for all acronyms and symbols needs to be more extensive. Not sure exactly how you defined 'lateral-averaged effectiveness' vs. 'spatially-averaged effectiveness'. [You defined eta_s on p.4, but never eta_l.] Eta_s equation should be better formatted (e.g., denominator 21x4 is confusing)
- Discuss the effect of pressure ratio across the hole. The original hole geometry is cylindrical, which naturally limits the speed of the flow everywhere, as does the flared-inlet geometry. But the flared outlet geometry could become supersonic (con-di), if the pressure ratio is high enough and flow heating occurs in the hole. Figures 16-18 should show the flow speed profile as Mach contours instead of dimensional speed, to illustrate how close this is to choking.
- Many typos need to be reviewed.
Author Response
Please see the detail of our answers in the attached file

Reviewer 2 Report
The paper presents a numerical study of film cooling performance and efficiency for a certain hole geometry that can be used in turbine blades. The authors use a commercial code (ANSYS CFX) to compute the 3-dimensional RANS turbulent flowfield through the cooling passage. It is clear from reading the manuscript that the paper is well-thought and executed, with sufficient initial validation & verification steps that examine the fit of the model parameters and the numerical grid. The results are clear and convincing, and not at all insignificant: an increase in cooling performance/efficiency by a few percentage points is quite important. The only issue that I see with the paper is the 'placement' of the present work with respect to past work. There is an extensive list of past works (1-20) most of which (7-10, 12-20) examine the effect of the hole size, shape, geometry, fluid mechanics in a similar configuration. The authors need to clearly distinguish what in the paper is their own contribution, and how their numerical analysis and results advance the state-of-the-art in the topic. A little more clarity is necessary with regards to what is new in the present work, even if it comes out to be - at the end - nothing more than a quantitative increase of cooling performance.
Author Response
Please see the detail of author answers in the attached file.
